# Lymphatic Complications Prevention and Soft Tissue Reconstruction after Soft Tissue Sarcoma Resection in the Limbs

**DOI:** 10.3390/medicina58010067

**Published:** 2022-01-02

**Authors:** Mario F. Scaglioni, Matteo Meroni, Elmar Fritsche, Bruno Fuchs

**Affiliations:** 1Clinic of Hand-and Plastic Surgery, Department of Surgery, Luzerner Kantonsspital, 6000 Lucerne, Switzerland; meroni369@gmail.com (M.M.); elmar.fritsche@luks.ch (E.F.); 2Clinic of Orthopedic Surgery, Department of Surgery, Luzerner Kantonsspital, 6000 Lucerne, Switzerland; bruno.fuchs@luks.ch

**Keywords:** lymph interpositional flap, lymphatic flow through flap, supermicrosurgery, lymphedema surgery, SCIP flap

## Abstract

*Background and Objectives:* The definitive treatment of soft tissue sarcomas (STS) requires a radical surgical removal of the tumor, which often leads to large soft tissue defects. When they are located in the limbs, significant damage to the lymphatic pathways is not uncommon. In the present article, we present different techniques aimed at both reconstructing the defect and restoring sufficient lymph drainage, thus preventing short- and long-term lymphatic complications. *Materials and Methods:* Between 2018 and 2020, 10 patients presenting a soft tissue defect with lymphatic impairment received a locoregional reconstruction by means of either pedicled or free SCIP flap. Seven patients required a second flap to reach a good dead space obliteration. In six cases, we performed an interpositional flap, namely a soft tissue transfer with lymphatic tissue preservation, and in four cases a lymphatic flow-through flap. In all cases, the cause of the defect was STS surgical excision. The average age was 60.5 years old (ranging 39–84), seven patients were females and six were males. *Results:* All the patients were successfully treated. In two cases, minor post-operative complications were encountered (infected seroma), which were conservatively managed. No secondary procedures were required. The average follow-up was 8.9 months (ranging 7–12 months). No signs of lymphedema were reported during this time. In all cases, complete range of motion (ROM) and a good cosmetic result were achieved. *Conclusions:* A reconstructive procedure that aims not only to restore the missing volume, but also the lymphatic drainage might successfully reduce the rate of postoperative complications. Both lymphatic interpositional flaps and lymphatic flow-through flaps could be effective, and the right choice must be done according to each patient’s needs.

## 1. Introduction

Soft tissue sarcomas are a wide range of malignant tumors that can present throughout the body. They are commonly located in the lower limbs, but in rare cases they might also be localized in the upper limbs [1]. Surgical resection with negative margins is considered the most effective treatment in terms of local recurrence-free, disease-specific and metastasis-free survival [2]. This kind of invasive procedure inevitably leads to significant tissue damage, and the remaining defects often need a reconstruction requiring soft tissue transfer. Depending on the affected area, the consequences of such an aggressive surgery might be related not only to missing volumes and shape, but also to functional issues. In particular, when a lymphatic-rich region is involved, complications such as lymphoceles or lymphedema are rather frequent if not properly treated [3].

These concerns were initially raised for breast cancer-related lymphedema, which is a rather common complication of axillary lymph node dissection [4,5]. Over the last few years, different solutions have been proposed to face lymphatic issues, but it is still widely debated which is the ideal treatment. Among them, the most valuable options described so far in terms of lymphatic flow restoration are lymphovenous anastomosis (LVA) and vascularized lymph node transfer (VLNT) [6,7]. However, both of them were intended to treat an already evident disease, and not to prevent its manifestation.

In the present work, we share our experience with slightly different procedures focused to fulfill the reconstructive requirements while restoring a sufficient lymphatic flow, in order to prevent short- and long-term sequelae. To achieve this, we resorted, in all cases, to the superficial circumflex iliac artery perforator (SCIP) flap, in either its pedicled or free form, and to its lymphatic network, which was used to perform lymphatic tissue transfer. Then, in selected cases we also exploited the flap’s vessels to perform a lymphatic flow-through (LyFT) flap, which consists of additional lymphovenous or lympho-lymphatic anastomoses (LLA) between the recipient site and the flap to enhance the immediate drainage potential.

## 2. Materials and Methods

In this retrospective study, 13 patients were included who underwent surgical resection of a soft tissue sarcoma requiring a soft tissue reconstruction (Table 1). These procedures were performed between 2018 and 2020 at the Luzerner Kantonsspital, Luzern, Switzerland by the same surgeon. Seven patients were females and six were males. The average age was 60.5 years old (ranging 39–84). The patients were divided into two groups according to the performed reconstructive procedure. Seven of them received a pedicled lymphatic interpositional flap, while six patients received a lymphatic flow-through flap (four pedicled and two free flaps). Among the lymphatic flow-through flaps, an average of 2.3 (range 1–3) lymphovenous bypasses were performed.

Indocyanine green (ICG) lymphography was executed, in all cases, preoperatively to identify the lymphatic pathway, and intraoperatively to visualize the leaking lymphatic vessels and to check the patency of the LVAs.

## 3. Results

All the patients received a superficial circumflex iliac artery perforator (SCIP) flap. For a better dead space obliteration, a second flap was performed in seven cases, among them four were deep inferior epigastric perforator (DIEP) flaps and three anterolateral thigh (ALT) flaps. No partial or complete flap losses were encountered. Two patients, one in each group, developed infected seroma, which were managed conservatively with fine-needle aspiration and supplemental antibiotic therapy. No secondary procedures were required.

The mean follow-up was 8.9 months (ranging 7–12 months). In all cases, complete range of motion (ROM) and a good cosmetic result were achieved. No signs of lymphocele or lymphedema were encountered. No signs of primary disease recurrency were reported during the follow-up.

### 3.1. Case 1

A 39-year-old male patient affected by an upper medial thigh sarcoma requiring surgical excision was referred to our department to plan a reconstructive solution. The preoperative ICG lymphography confirmed the involvement of many lymphatic vessels, therefore we had the goal of filling the defect while restoring the lymph drainage. Because of its proximity and good tissue quality, we chose a pedicled SCIP flap, which also contained some large lymphatics (Figure 1A). The sarcoma was surgically removed leaving a 16 cm × 16 cm defect (Figure 1B). The pedicled SCIP flap was then harvested, supplied by the superficial branch of the superficial circumflex iliac artery, and the superficial circumflex iliac vein was isolated (Figure 1C). The flap was then rotated by 180° to reach the recipient site through an inguinal tunnel and the vein anastomosed to a branch of the superficial femoral vein (Figure 1D). A critical point at this stage is to carefully orient the direction of the transposed lymphatic vessels, matching the direction of the native ones. The flap was partially de-epithelized and buried to fill the defect, leaving a skin island to monitor the perfusion. Since extensive damage of the native lymphatics was inevitable during the sarcoma excision, additional lymphovenous anastomoses were performed at the distal margin with the intention of reducing the lymph flow in the affected area immediately after surgery (Figure 2). The postoperative course was uneventful, no complications were reported, and no secondary procedures were required. At the 10-month follow-up, the flap had completely healed with a good aesthetic result, and no signs of lymphedema were noted (Figure 3).

### 3.2. Case 2

A 66-year-old female patient affected by a forearm soft tissue sarcoma requiring a radical surgical excision was referred to our department to plan a reconstructive solution. We performed pre-operative indocyanine green lymphography that showed a very rich lymphatic network running into the affected area (Figure 4A). The tumor was excised leaving a defect of 7 cm × 9 cm with muscle exposure. Considering the extension of the defect and the need for a thin and pliable tissue, we chose a free SCIP flap. We extended the lymphatic analysis to the donor site and marked the route of the vessels (Figure 4B). This helped us to identify and preserve a large lymphatic vessel suitable for a lympho-lymphatic anastomosis. Two leaking lymphatic vessels were also prepared at the distal margin of the recipient site (Figure 4C). The free flap was then anastomosed and inset, carefully orienting the direction of the transposed vessel in order to match the native ones (Figure 4D). A lymphovenous and lympho-lymphatic anastomoses were then performed. Patency of both the anastomoses was checked by means of intraoperative ICG imaging (Figure 5). The post-operative course was uneventful, and no complications were reported. At the 12-month follow-up, the flap had completely healed with a good cosmetic result and no signs of lymph stasis were noted (Figure 6).

## 4. Discussion

Sarcomas are a subtle category of tumors that, depending on their location, may remain silent until reaching a significative volume. If diagnosed promptly, a safe margin resection may result in definitive treatment [8]. This, however, often implies a significative remaining defect. Even more massive defects are the results of resections of tumors that were initially treated conservatively. This occurs rather frequently in older patients in delicate general conditions, but the growth of the tumor may result in a very distressing mass effect making surgery inevitable. Every time such massive resections are performed, the result is severe soft tissue damage and, depending on the location of the affected site, they can cause a compromise of the lymphatic drainage pathway [9]. This is particularly relevant in the upper thigh region, but also in the upper limbs large resections have severe consequences in terms of lymph stasis.

Nowadays, from the reconstructive point of view disparate options are available in the plastic surgeon armamentarium, especially with the advent of perforator-based flaps. They offer a series of advantages ranging from shorter operative time to less donor site morbidity and are becoming more and more reliable throughout the body [10,11]. In particular, the SCIP flap is one of the most promising techniques. It is relatively simple and quick to harvest, its thickness and extension can be tailored according to need, it is very pliable, the donor site morbidity is low, and the aesthetic result is good, with the remaining scar relatively easy to hide [12]. Even chimeric forms are possible, including multiple skin islands, bone, muscle, nerve and lymph nodes [13].

From the lymphatic point of view, a strong debate is still ongoing. The development of postoperative lymphedema brought the attention towards the need of a preventive treatment for lymphatic problems. At the moment, different procedures are available to reduce the severity of the disease, but the most accepted procedure in terms of prevention remains only the immediate execution of LVAs. Mainly described as LYMPHA (Lymphatic Microsurgical Preventive Healing Approach), it consists of performing multiple lymphovenous bypasses at the time of surgery [14]. In the last few years, other procedures have been proposed in order to exploit the potential of the transferred tissue to prevent these complications. Among them, two options seem to be more promising: the use of lymphatic-rich tissues as interpositional flaps and the lymphatic flow-through (LyFT) flaps.

The lymphatic interpositional flaps consist of preserving the lymphatic vessels running into the donor tissue and moving them into the affected area in order to replace the missing ones. It is a recent and fascinating procedure which relies on the neolymphangiogenesis process stimulated by the donor vessels to regenerate a physiological draining pathway. Moreover, this type of tissue presents a good potential in terms of fluid absorption that may be useful immediately after surgery. Therefore, every time a soft tissue transfer is needed to fill a defect, this technique may be attractive because of its simpleness. As suggested by Yamamoto et al., the orientation of the tissues is of critical importance in order to obtain a satisfactory result [15]. Despite the novelty of the procedure, studies performed employing muscle and skin flap have demonstrated this process [16,17], showing both a spontaneous reconnection of the recipient site lymphatic vessels with the donor ones and an increased neolymphangiogenesis [18]. A limitation of this procedure is that it requires some time to actually be effective. For this reason, to prevent major lymphatic damage, a safe approach should also include a combination with LVAs.

In this respect, the lymphatic flow-through flap (LyFT) might be the ideal solution. Initially proposed by Fujiki et al. to reconstruct arterial and venous defects [19], it was then proposed by di Summa and Guiller to restore the lymphatic drainage [20]. This procedure consists of performing lymphovenous anastomoses between leaking damaged lymphatic vessels and superficial veins of the flap. In the present work, we show a further evolution of this procedure by also performing a lympho-lymphatic anastomosis between a recipient site interrupted vessel and a flap’s lymphatic vessel. Even if more technically demanding, this allows the surgeons to combine the long-term effects of the interpositional flaps with the immediate flow restoration of LVAs, which is particularly relevant in terms of peri-operative complications prevention (such as lymphocele).

## 5. Conclusions

Lymphatic complications are complex and multi-faceted issues that require a treatment tailored according to the patient’s need. For this reason, despite the limited evidence available so far, we believe that the lymphatic interpositional flaps and the lymphatic flow-through flaps are two modern and valuable solutions to take into account regarding the reconstruction of the large defects typical of sarcoma resection surgeries.

## Figures and Tables

**Figure 1 medicina-58-00067-f001:**
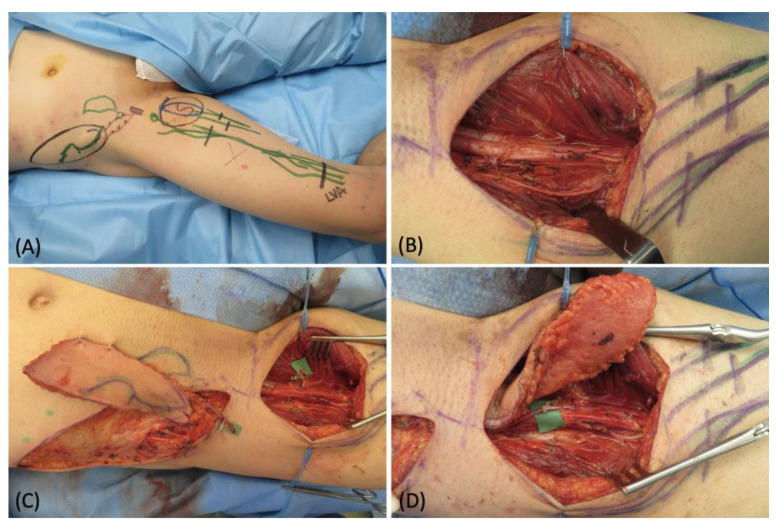
(**A**) Pre-operative picture with skin markings after ICG lymphography; (**B**) surgical field after tumor removal; (**C**) SCIP flap harvest with superficial circumflex iliac vein isolation; (**D**) flap inset and venous pedicle anastomosis.

**Figure 2 medicina-58-00067-f002:**
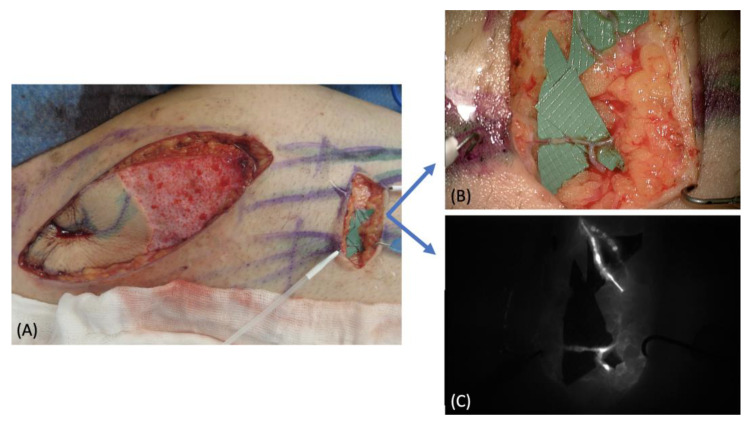
(**A**) Partial flap de-epithelization and preparation of lymphatic vessels for the anastomoses; (**B**) intraoperative picture of the LVAs under the microscope; (**C**) ICG imaging to check the patency of the anastomoses.

**Figure 3 medicina-58-00067-f003:**
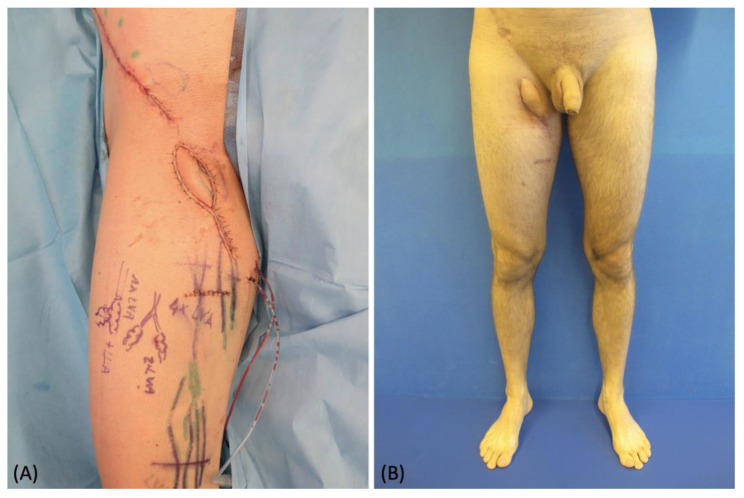
(**A**) Intraoperative picture at the immediate end of the procedure; (**B**) post-operative picture at 10-month follow-up.

**Figure 4 medicina-58-00067-f004:**
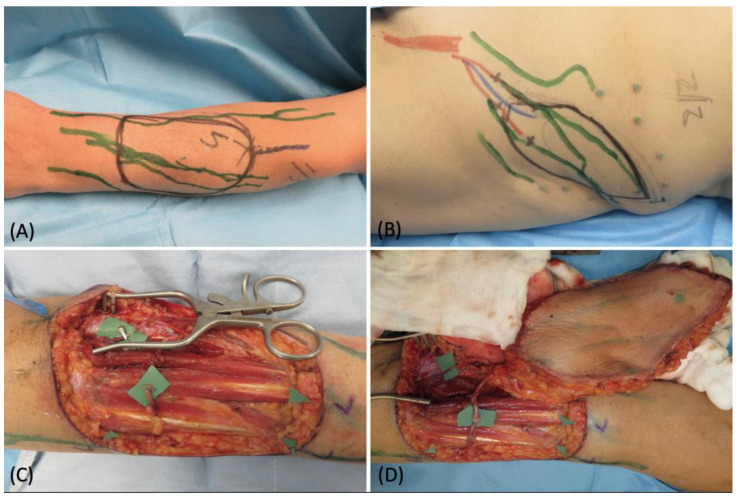
(**A**) Recipient site pre-operative picture with skin markings after ICG lymphography; (**B**) donor site pre-operative picture with skin markings after ICG lymphography; (**C**) surgical field after tumor removal; (**D**) free flap transposition with its arterial and venous anastomoses.

**Figure 5 medicina-58-00067-f005:**
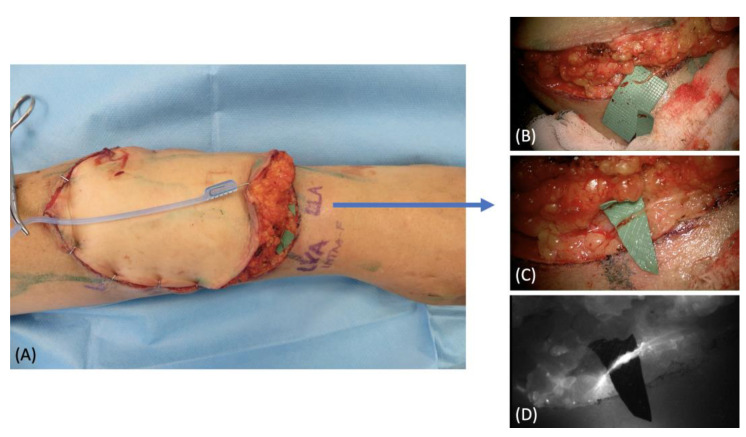
(**A**) Preparation of the lymphatics and veins for the anastomoses; (**B**) intraoperative picture under the microscope of the lympho-lymphatic anastomosis preparation; (**C**) picture after the anastomosis; (**D**) ICG imaging to check the patency.

**Figure 6 medicina-58-00067-f006:**
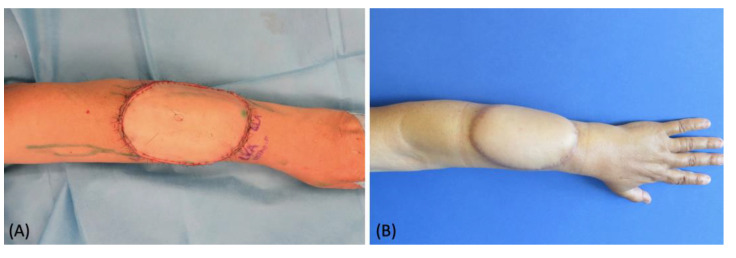
(**A**) Intraoperative picture at the immediate end of the procedure; (**B**) post-operative picture at 10-month follow-up.

**Table 1 medicina-58-00067-t001:** Patient demographic and case characteristics.

	Patient Number	Gender	Mean Age	Type of Flap	Complications	Mean Follow-Up Time	Outcomes
**Lymphatic interpositional flap**	7	3 M; 4 F	62.8 y.o. (range 39–84)	7 pedicled 0 free	1 infected seroma	9.8 months (range 7–12)	Full ROM No lymphocele/lymphedema
**Lymphatic flow-through flaps**	6	3 M; 3 F	57.6 y.o. (range 42–76)	4 pedicled 2 free	1 infected seroma	7.8 months (range 9–12)	Full ROM No lymphocele/lymphedema

## Data Availability

Data are available from the authors upon reasonable request.

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
