# Peer review of "Lymphatic Complications Prevention and Soft Tissue Reconstruction after Soft Tissue Sarcoma Resection in the Limbs"

_medicina, 2022, doi:10.3390/medicina58010067_

Round 1

Reviewer 1 Report

This is an interesting work facing immediate reconstructin of the lymphatic flow after surgery of sarcomas. Tha authors demonstrated the new establishment of lymphatic circulation among flaps, which served to improve it and avoided the production of lymphedema. The reconstructive technique is amazing and the authors showed a huge experience in this field.  Nevertheless it will be necessary to repeat those protocols with a higher number of patients, to confirm the findings.

Author Response

we thank the reviewer for the comments. 

We totally agree with him, these are preliminary results.

we are working on producing a large  series of cases.

Thank you very much indeed for your positive feedback.

Reviewer 2 Report

As we all know, lymphatic complications after tumor removal may be dramatic. The manuscript presents a novel and effective technique in the reconstruction of soft tissue with immediate lymphatic reconstruction. This operational model was presented in 10 cases, showing the effectiveness and meaningfulness of the method.
I have found the submission very interesting.

I hope the authors will continue the technique to show a larger patient volume and longer follow-up in the next manuscript.

Author Response

I thank the authors for their positive feedback.

We are going to continue with this technique as we are obtaining encouragement results. 

Thank you for supporting our work.

Reviewer 3 Report

First of all, congratulations to the authors for their research. Preventative restoration of the lymphatic flow in reconstructive soft-tissue surgery is a novel and laudable approach. 

A few comments, though:

  1. Ideally, the study should have been a prospective one. It is not a significant problem that the authors propose a retrospective study, but the follow-up period should have been longer than 12 months. The authors state that the surgeries took place between 2018 and 2020. What happened with the patients operated on in 2018? Why weren't they followed up longer? The authors should mention their reasons for choosing a 12-month follow-up time frame.
  2. The authors should expand the contents of table 1 and add a thorough description for every patient, including diagnosis and complete surgical plan. Since the study only included thirteen patients and the primary illness is cancer, every patient-related detail is essential.
  3. Since the article's focus is lymphatic reconstruction, a detailed description of the surgical technique would be a welcome addition.
  4. The authors should professionally edit the paper for the English language.

Author Response

Dear Reviewer,

thank you for your appreciation for our work and for your comments, here following are our answers:

  1. The follow up reported in the manuscript is limited to 12 months because we usually follow the patients in person for 1 year, then the following follow up visits are organized with oncologists and general surgeons, which refer to us in case of complications onset.
  2. We provide the details about the defects and the surgical plans in both Patients and Methods, and Results section. For conciseness reasons, and according to the guidelines, we prefer to describe only a summary of all the patients’ data in Table 1.
  3. We did not provide a specific surgical technique section because it is not the focus of the present work. All the employed procedures have been already described either by us or other authors in specific manuscripts. With the present article we would like to gather our experience with these procedures, showing their efficacy in case of limb’s sarcoma resections.
  4. The manuscript has been completely revised by native english speaker.